# New Cysteine-Containing PEG-Glycerolipid Increases the Bloodstream Circulation Time of Upconverting Nanoparticles

**DOI:** 10.3390/molecules27092763

**Published:** 2022-04-26

**Authors:** Maria E. Nikolaeva, Andrey V. Nechaev, Elena V. Shmendel, Roman A. Akasov, Mikhail A. Maslov, Andrey F. Mironov

**Affiliations:** 1Lomonosov Institute of Fine Chemical Technologies, MIREA—Russian Technological University, 86 Vernadsky Ave., 119571 Moscow, Russia; chemorg@mail.ru (A.V.N.); elena_shmendel@mail.ru (E.V.S.); mironov@mitht.ru (A.F.M.); 2Federal Scientific Research Centre “Crystallography and Photonics” of RAS, 59 Leninsky Ave., 119333 Moscow, Russia; roman.akasov@gmail.com

**Keywords:** upconverting nanoparticles, PEG-lipids, surface modification

## Abstract

Upconverting nanoparticles have unique spectral and photophysical properties that make them suitable for development of theranostics for imaging and treating large and deep-seated tumors. Nanoparticles based on NaYF_4_ crystals doped with lanthanides Yb^3+^ and Er^3+^ were obtained by the high-temperature decomposition of trifluoroacetates in oleic acid and 1-octadecene. Such particles have pronounced hydrophobic properties. Therefore, to obtain stable dispersions in aqueous media for the study of their properties in vivo and in vitro, the polyethylene glycol (PEG)-glycerolipids of various structures were obtained. To increase the circulation time of PEG-lipid coated nanoparticles in the bloodstream, long-chain substituents are needed to be attached to the glycerol backbone using ether bonds. To prevent nanoparticle aggregation, an L-cysteine-derived negatively charged carboxy group should be included in the lipid molecule.

## 1. Introduction

Methods of drug delivery are currently a topic of intense research. Liposomes, nanocapsules, and multifunctional nanoparticles are among the most well-known delivery systems [1,2,3,4,5,6,7]. An important feature of these systems is the possibility of targeted delivery to specific tissues and cells by modifying the surface of the carrier with molecules that bind to receptors on the target cells [5,8,9]. As a result, particles are preferentially internalized by specific cells, which allows for a significant reduction in drug therapeutic dose, frequency of treatment, and side effects.

Colloidal inorganic nanocrystals are of great interest because they have unique properties depending on particle size and shape. Nanophosphors are excited by light in the near-infrared range, which can penetrate biological tissue to a depth of several centimeters. The luminescence spectrum of upconverting nanoparticles has intense bands in the visible and UV regions, nanoparticles can transfer energy to photosensitizers without emission, and they have high photostability [10,11,12,13,14,15,16,17].

Since the synthesis of upconverting nanoparticles is performed in high-boiling-point solvents oleic acid and 1-octadecene, the carboxyl groups of oleic acid form coordination bonds with the electron-deficient metal surface during synthesis, resulting in hydrophobic nanoparticles. Therefore, a PEG-lipid decoration approach was used to modify the surface of the nanoparticles to make them more biocompatible. PEG-containing glycerolipids could solubilize hydrophobic upconverting nanoparticles in biological media and to protect from exposure to components of serum, which leads to increased circulation time in the bloodstream [13].

## 2. Results and Discussion

In the present work, we performed the surface modification of hydrophobic upconverting nanoparticles using different PEG-lipids (Figure 1) to evaluate how their structure affects the physicochemical and biological properties, and bloodstream circulation time of upconverting nanoparticles. 

PEG-lipid **1** is commercially available and used to modify the surface of nanoparticles used in the photodynamic therapy of skin cancer [14] and for the stabilization of colloidal nanocomposites [15]. However, hydrocarbon chains in lipid **1** are attached to the glycerol backbone via ester bonds, whereas in the new lipids, **2**–**4** hydrocarbon chains are attached via ether bonds. Since ester bonds are less susceptible to hydrolysis by esterases, this should increase the retention time of nanoparticles in the bloodstream.

For the synthesis of PEG-lipid, **2** *rac*-1-*O*-(4-nitrophenyloxycarbonyl)-2,3-di-*O*-tetradecylglycerol **5** [16,17] was treated with amino-methoxy-PEG_2000_ resulting in the formation of product **2** with a 42% yield (Figure 1).

PEG-lipid **3** was synthesized from *rac-*1,2-di-*O*-octadecylglycerol **6**, which was treated with imidazole complex with phosphorus (III) chloride (Figure 2). Obtained product **7** was then activated with pivaloyl chloride and condensed with methoxy-PEG_2000_ following by oxidation with iodine. After isolation by means of column chromatography, compound **3** was obtained with 38% yield over the two stages. Characteristic proton signals of glycerol CH_2_O-group (dd, 4.22–4.10 ppm) and PEG residue (br.s, 3.65 ppm) were found in the ^1^H NMR spectrum of lipid **3**.

For synthesis of PEG-lipid **4** containing a cysteine residue, the initial *rac-*1,2-di-*O*-octadecylglycerol **6** was treated with *N*-bromosuccinimide, and the resulting bromo derivative **8** [18] was condensed with Boc-L-cysteine methyl ester in the presence of a catalytic amount of KI (Figure 2), giving compound **9** with 81% yield. Signals of L-cysteine protons (dt, 4.40 ppm, *J* 5.5, 7.9 Hz, CH-Cys and m, 2.85–2.81 ppm, SCH_2_-Cys) and Boc-group (s, 1.41 ppm С(Me)_3_) were found the ^1^H NMR spectra. Treatment of compound **9** with HCl/MeOH gave product **10**, which was condensed with succinic anhydride to give compound **11** with 88% yield. Activation of the carboxyl group of compound **11** with *N*-hydroxysuccinimide and subsequent condensation with amino-methoxy-PEG_2000_ gave PEG-lipid **12** with 63% yield. In the ^1^H NMR spectra of compounds **12,** the proton signals of PEG residue, and L-cysteine and glycerol (δ 3.90–3.41 ppm, CH_2_O, 2 OCH_2_CH_2_, CH_3_O-Cys, PEG, CH_2_S) confirmed the formation of the desired product. To obtain the final ionic form of lipid **4**, compound **12** was treated successively with lithium hydroxide and sodium hydrogen carbonate, which produced lipid **4** with 61% yield over the two stages.

Upconverting nanoparticles can be produced by various methods [19,20]. We developed a method of synthesis of upconverting nanoparticles that renders possible to obtain monodisperse particles of NaYF_4_:M (М = Yb^3+^:Er^3+^) with a high conversion yield [21]. Oxides Y_2_O_3_, Yb_2_O_3_, and Er_2_O_3_ were boiled together in TFA-H_2_O (3:1 vol.); then, CF_3_COONa was added and heated in a mixture of high-boiling-point solvents 1-octadecene and oleic acid (Figure 2A). A special feature of this synthesis is a rapid temperature increase at the beginning of synthesis (50 °C per min) and a sharp temperature decrease (100 °C per min) at the end of the synthesis to obtain monodisperse nanoparticles with a particle size of 75 ± 5 nm (Figure 2B). The synthesis allows for obtaining particles with a conversion coefficient (ratio of radiated power to absorbed power) of 14% at an excitation radiation intensity of 10 W/cm^2^.

Nanoparticles are rapidly removed from the bloodstream due to the reticuloendothelial system, followed by their accumulation in the spleen, kidneys, and liver [8]. Particles coated with different shells revealed reduced adsorption by blood proteins and phagocytosis by macrophages, and are characterized by prolonged circulation time in the organism [22]. To modify the surface of the upconverting nanoparticles, a solution of PEG-glycerolipids **1**–**4** in dry chloroform was added to the nanoparticles. Chloroform was evaporated; a thin film was dried under 0.1 torr for 3 h and then suspended in water (MilliQ) to give a dispersion that was passed through a 0.2 μm PTFE membrane filter (Merck, Darmstadt, Germany). The dispersion was centrifuged at 6000× *g* for 1 h, and nanoparticles modified with PEG-lipids **1**–**4** were redispersed in sterile water. Slightly opalescent dispersions were formed in the case of PEG-lipids **1**, **3** and **4**, which have ionizable groups, whereas in the case of the uncharged lipid **2** a nontransparent rapidly settling aqueous dispersion was obtained. Particle size and zeta potential were evaluated using a Zetasizer Nano ZS (Malvern Panalytica, Malvern, UK) at 25 °C in deionized water (size) or in 10 mM NaCl solution (zeta-potential). Upconverting nanoparticles coated by PEG-lipid **4** had a size of 165 ± 62 nm and zeta potential of −23.1 mV (see Appendix A).

To study the circulation time of upconverting nanoparticles modified with PEG-glycerolipids **1**–**4** in the bloodstream, a series of in vivo experiments were performed. The mouse line of BDF1 (C57Bl/6 × DBA2) was anaesthetized by intraperitoneal injection of 0.2 mL zoletil-rometar. Nanoparticles were injected (0.15 mL) intravenously at a dose of 5 mg/kg. Blood sampling was performed from the tail vein of the mice after 1, 3, 5, 10, 30, 60, 120, and 180 min, and 1 day with tail clipping. Each blood sample was examined on an upconverting fluorescence microscope at four locations (Figure 2C). The upconverting signal in blood for particles coated with PEG-lipid **1** was visible up to 15 min, and for PEG-lipid **2** up to 5 min. The signal from the nanoparticles modified by ionizable lipid **3** was observed up to 1.5 h, whereas nanoparticles modified with cysteine-containing PEG-lipid **4** were visible for more than 3 h, more than 15-fold longer than the result obtained for the commercially available PEG-lipid **1**. In addition, the circulation time of nanoparticles coated with PEG-lipid **4** was more than 3-fold longer than the time of other upconverting nanoparticles with different shells described in the literature [23,24]. Analysis of the microphotographs using ImageJ (NIH, Bethesda, MD, USA) software showed that the nanoparticle number was reduced from 260 to 18 over 10 min (Figure 2D), whereas only a moderate decrease in fluorescence intensity was observed (Figure 2E). 

Nanoparticles coated with PEG-lipid **4** were nontoxic for Bj-5ta fibroblasts after 24 h incubation (Figure 2F), and cell death measured by MTT assay did not exceed 10%. Thus, these upconverting nanoparticles were well-tolerated by the cells and could be considered to be promising candidates for in vivo applications.

## 3. Materials and Methods

### 3.1. General Methods

HPLC-Kieselgel plates were used for thin-layer chromatography 60 F_254_ (Merck, Darmstadt, Germany). Column chromatography was performed on Silicagel 60 (40–63 μm; Merck, Darmstadt, Germany). Resines Dowex 50WX2 Н^+^ and Dowex 50WX8 H^+^ were used for ion-exchange chromatography. NMR spectra were obtained on a DPX-300 spectrophotometer (Bruker, Billerica, MA, USA) with an operating frequency of 300 MHz. Mass spectra were run on a Ultraflex time-of-flight mass spectrometer (Bruker, Billerica, MA, USA) with matrix-assisted laser desorption/ionization (MALDI). High-resolution TEM studies were performed using a JEM-ARM200F (Jeol, Tokyo, Japan) cold FEG double aberration corrected electron microscope operated at 200 kV and equipped with a large solid-angle Centurio EDX detector and Quantum EELS spectrometer. The detection of the photoluminescent signal was performed using Falcon EMCCD camera (Raptor Photonics, Milbrook, UK), equipped with the F = 0.95 objective.

### 3.2. Synthesis of PEG-Glycerolipids

*rac*-1-*О*-[Methoxy(polyethylene glycol-2000)aminocarbonyl]-2,3-di-*O*-tetradecylglycerol (**2**). Compound **5** (213 mg, 0.328 mmol) was dissolved in 3 mL of anhydrous DCM, then 550 mg (0.275 mmol) of amino-methoxy-PEG_2000_ and 150 μl anhydrous TEA was added. The reaction mixture was stirred for 24 h at 24 °C. The product was purified by ion-exchange chromatography on resin Dowex 50WX8 H^+^ (20% aq. MeOH) to give 286.3 mg (42%) of PEG-glycerolipids **2**. Mass spectrum (MALDI) *m/z*: [M+Na]^+^: 2501.921 (major peak).^1^H-NMR (CDCl_3_), δ: 5.30–5.38 (m, 1 H, NHCH_2_), 4.13–4.22 (m, 1 H) and 4.09 (dd, *J* = 5.5, 11.5 Hz, 1 H, CH_2_OC(O)), 3.75–3.80 (m, 1 H, OCH_2_CHO), 3.31–3.75 (m, 185 H, NHCH_2_, CH_2_OCH_3_, PEG-H, 2 OCH_2_CH_2_, CH_2_OCH_2_), 1.51–1.63 (m, 4 H, 2 OCH_2_CH_2_), 1.18–1.38 (m, 44 H, 2 (CH_2_)_11_CH_3_), 0.87 (t, *J* = 6.9 Hz, 6 H, 2 (CH_2_)_11_CH_3_).

*rac*-1,2-Di-*O*-octadecyl-3-*H*-phosphoglycerate (**7**). A solution of phosphorus (III) chloride (56 μL, 0.54 mmol) in anhydrous DCM (20 mL) was added dropwise to a solution of imidazole (200 mg, 2.52 mmol) in DCM. TEA (234 μL, 1.44 mmol) was added to the solution, and the reaction mixture was stirred for 15 min. Compound **6** (700 mg, 0.18 mmol) was dissolved in DCM added to the mixture, and stirred at 24 °C. After 1.5 h 1N aq. HCl (50 mL) was added, and compound **7** was extracted with chloroform, washed with water, and dried over sodium sulfate. After purification by column chromatography on silica gel (chloroform-methanol, 100:1), compound **7** was obtained in 51% yield (388 mg). ^1^Н-NMR (CDCl_3_), δ: 8.26 (d, *J* = 2.9 Hz, 1 H, ОН), 4.77 (dd, *J* = 2.7, 14.3 Hz, 1 H, H_2_PO_3_), 4.13–4.22 (m, 1 H) and 4.04 (dd, *J* = 5.0, 7.1 Hz, 1 H, OCH_2_CHO), 3.94–3.86 (m, 1 H, OCH_2_CHO), 3.58–3.72 (m, 2 H, CH_2_OCH_2_), 3.37–3.51 (m, 4 H, 2 OCH_2_CH_2_), 1.45–1.61 (m, 4 H, 2 OCH_2_CH_2_), 1.20–1.37 (m, 60 H, 2 (CH_2_)_15_CH_3_), 0.89 (t, *J* = 6.9 Hz, 6 H, 2 (CH_2_)_15_CH_3_).

1,2-Di-*O*-octadecyl-*rac*-glycero-3-(methoxy(polyethylene glycol-2000)phosphate (**3**). Compound **7** (300 mg, 0.455 mmol) was suspended in pyridine, and 300 μL (4.57 mmol) of pivaloyl chloride and 910 mg (0.455 mmol) of amino-methoxy-PEG_2000_ were added. Reaction mixture was stirred for 1.5 h when heated to 50 °C, and 231 μL I_2_ was dissolved in a mixture pyridine-H_2_O (98:2 vol.) was added to the reaction flask. After 1.5 h 5% aq. sodium hydrosulfite was added and the compound **3** was extracted with chloroform, washed with water, dried with sodium sulfate. PEG-glycerolipids **3** was purified by column chromatography on silica gel in chloroform-methanol (15:1) to give 710 mg (59% yield). ^1^Н-NMR (300 MHz, CDCl_3_), δ: 4.13–4.22 (m, 1 H) and 4.10 (dd, *J* = 5.0, 7.1 Hz, 1 H, OCH_2_CHO), 3.98–3.91 (m, 1 H, OCH_2_CHO), 3.18–3.72 (m, 185 H, CH_2_OCH_2_, CH_2_OCH_3_, PEG-H, 2 OCH_2_CH_2_), 1.59–1.49 (m, 4 H, 2 OCH_2_CH_2_), 1.30–1.22 (m, 60 H, 2 (CH_2_)_15_CH_3_), 0.87 (t, *J* = 7.0 Hz, 6 H, 2(CH_2_)_15_CH_3_).

*S*-(1,2-Di-*O*-octadecyl-*rac*-glycero)-*O*-methyl-*N*-(*tret*-butoxycarbonyl)-L-cysteine (**9**). Compound **8** (2.25 g, 3.41 mmol) was dissolved in DMFA (30 mL), and 3.7 g (27.28 mmol) of DIPEA, 6.8 g (27.28 mmol) of methyl *tret*-butoxycarbonyl-L-cysteine ester, and catalytic amount of potassium iodide were added. A reaction mixture was stirred for 4 h at 80 °C. The compound **9** was extracted with chloroform, washed with water and 0.1N aq. HCl, then was dried over sodium sulfate. Purification by column chromatography on silica gel (hexane-ethyl acetate, 20:1) gave 2.5 g (81%) of compound **9**. ^1^H-NMR (CDCl_3_), δ: 5.29–5.25 (m, 1 H, NH), 4.40 (dt, *J* = 5.5, 7.9 Hz, 1 H, CH-Cys), 4.12–3.40 (m, 12 H, 2 OCH_2_CH_2_, OCH_2_, CHCH_2_S, OCH_3_-Cys), 3.07 (dd, *J* = 5.3, 13.9 Hz, 1 H) and 2.85–2.81 (m, 1H, SCH_2_), 1.60–1.47 (m, 4 H, 2 OCH_2_CH_2_), 1.41 (s, 9 H, С(СH_3_)_3_), 1.36–1.20 (m, 60 H, 2 (CH_2_)_15_CH_3_), 0.91 (t, *J* = 6.9 Hz, 6 H, 2 (CH_2_)_15_CH_3_).

*S*-(1,2-Di-*O*-octadecyl-*rac*-glycero)-*O*-methyl-L-cysteine hydrochloride (**10**). To a solution of compound **9** (2 g, 2.46 mmol) in DCM 4*N* HCl/MeOH (2 g, 28 mmol) was added. After 12 h at 24 °C organic solvents were evaporated and the residue was resuspended in diethyl ether (40 mL), the obtained amorphous solid was filtered to give 1.76 g (88%) of compound **10**. ^1^H-NMR (CDCl_3_), δ: 3.96–3.85 (m, 1 H, CH-Cys), 3.40–4.12 (m, 12 H, 2 OCH_2_CH_2_, OCH_2_, OCH_3_-Cys, CHCH_2_S), 3.03 (dd, *J* = 4.6, 14.5 Hz, 1 H) and 2.82 (dd, *J* = 4.6, 14.5 Hz, 1 H, SCH_2_), 1.58–1.41 (m, 4 H, 2 OCH_2_CH_2_), 1.36–1.19 (m, 60 H, 2 (CH_2_)_15_CH_3_), 0.89 (t, *J* = 6.9 Hz, 6 H, 2 (CH_2_)_15_CH_3_).

*S*-(1,2-Di-*O*-octadecyl-*rac*-glycero)-*O*-methyl-*N*-[(2-carboxyethyl)carbonyl]-L-cysteine (**11**). Compound **10** (200 mg, 0.28 mmol) was dissolved in DCM and 15 mg (0.7 mmol) of DIPEA, 12.5 mg (0.7 mmol) of succinic anhydride were added. The mixture was stirred for 12 h at 24 °C, washed with water and 0.1N aq. HCl, dried over sodium sulfate, organic solvents were evaporated. Compound **11** (200 mg, 88%) was used without further purification.

*S*-(1,2-Di-*O*-octadecyl-*rac*-glycero)-*O*-methyl-*N*-[2-[methoxy(polyethylene glycol-2000)aminocarbonyl]ethylcarbonyl]-L-cysteine (**12**). Compound **11** (100 mg, 0.12 mmol) was dissolved in DCM, then 16 mg (0.13 mmol) of N-hydroxysuccinimide and 26 mg (0.13 mmol) of EDC were added. The mixture was stirred for 12 h at 24 °C, washed with water, dried over sodium sulfate and evaporated. The residue was dissolved in DCM, then 30 mg (0.22 mmol) of DIPEA and 440 mg (0.22 mmol) of amino-methoxy-PEG2000 were added. The mixture was stirred for 24 h at 24 °C, washed with water and 0.1N aq. HCl, dried over sodium sulfate and organic solvents were evaporated. The product was purified by column chromatography on silica gel (chloroform-methanol, 8:1) to give 213 mg (63%) of compound **12**. ^1^H-NMR (CDCl_3_), δ: 6.93–6.85 (m, 1 H, NH), 5.96–5.87 (m, 1 Н, NH-PEG), 4.30 (dt, *J* = 5.6, 8.6 Hz, 1 H, CH-Cys), 3.90–3.41 (m, 207 H, CH_2_O, 2 OCH_2_CH_2_, OCH_3_-Cys, CH_2_OCH_3_, PEG-H, CHCH_2_S), 3.02 (dd, *J* = 5.6, 13.8 Hz, 1 H) and 2.77 (dd, *J* = 5.6 Hz, 1 H, SCH_2_), 2.60–2.45 (m, 4 H, C(O)CH_2_CH_2_C(O)), 1.57–1.45 (m, 4 H, 2 OCH_2_CH_2_), 1.31–1.22 (m, 60 H, 2 (CH_2_)_15_CH_3_), 0.87 (t, *J* = 6.9 Hz, 6 H, 2 (CH_2_)_15_CH_3_)1.

*S*-(1,2-Di-*O*-octadecyl-*rac*-glycero)-*N*-[2-[methoxy(polyethylene glycol-2000)aminocarbonyl]ethylcarbonyl]-L-cysteine sodium salt (**4**). Compound **12** (100 mg, 0.04 mmol) was dissolved in 2.5 mL of a mixture THF–H_2_O (1:1, vol.) then 10% aq. solution of lithium hydroxide was added. The mixture was stirred for 2 h at 24 °C followed by acidification with aq. HCl to pH 4 and the product was extracted with DCM, dried over sodium sulfate and evaporated. The product was dissolved in a mixture MeOH–H_2_O (1:1 vol.) then 10% aq. solution of sodium hydrogen carbonate was added and stirred for 15 min. Solvents were evaporated, the residue was dried under 0.1 torr for 30 min and was resuspended in diethyl ether. The obtained amorphous solid was filtrated to give 99 mg (97%) of lipid **4**. ^1^H-NMR (CDCl_3_), δ: 6.93–6.87 (m, 1 H, NH), 5.96–5.90 (m, 1 Н, NH-PEG), 3.90–3.41 (m, 204 H, CH_2_O, 2 OCH_2_CH_2_, CH_2_OCH_3_, PEG-H, CHCH_2_S), 3.00 (dd, *J* = 5.6, 13.8 Hz, 1 H, SCH_2_), and 2.77 (m, 1 H, SCH_2_), 2.64–2.47 (m, 4 H, C(O)CH_2_CH_2_C(O)), 1.57–1.40 (m, 4 H, 2 OCH_2_CH_2_), 1.31–1.23 (m, 60 H, 2 (CH_2_)_15_CH_3_), 0.87 (t, *J* = 6.9 Hz, 6 H, 2 (CH_2_)_15_CH_3_).

### 3.3. Synthesis of Upconverting Nanoparticles NaYF_4_: Yb^3+^, Er^3+^

OxidesY_2_O_3_, Yb_2_O_3_, Er_2_O_3_ were boiled in the mixture TFA–H_2_O (3:1 vol.) until dissolved. The solvents were slowly evaporated. To obtain the β-hexagonal phase CF_3_COONa (2 eq.) 1-octadecene (15 ml) and oleic acid (15 mL) were added to the trifluoroacetates. The mixture was heated to 100 °C for 30 min and then to 180 °C in an argon atmosphere. To decompose the trifluoroacetates the flask was placed in a Rose alloy heated to 360 °C. After 25 min the flask was removed from the Rose alloy and 1-octadecene (15 mL) was added for rapid cooling. The upconverting nanoparticles were washed with isopropanol and centrifuged at 6000 rpm for 30 min (Z206A centrifuge, Hermle, Germany).

### 3.4. Modification of Upconverting Nanoparticles with the PEG-Glycerolipids **1, 2, 3, 4**

PEG-lipids **1**–**4** (16 mg) was dissolved in chloroform and mixed with upconverting nanoparticles (4 mg) dispersed in 4 ml of chloroform (particle concentration 1 mg/mL). The resulting mixture was evaporated to form a thin film and was dried under 0.1 torr for 3 h. The particles were suspended in water and filtered through a 0.2 μm PTFE filter (Merck, Darmstadt, Germany), then were centrifuged for 1 h at 6000 rpm (Z206A centrifuge, Hermle, Germany). Then the modified particles were dispersed in 2 ml of 0.9% aq. sodium chloride solution.

Used a model of black mice with the introduction of anesthesia zoletil-rometar 0.2 mL intraperitoneally in the amount of 3 pieces. A conjugate of PEG-glycerolipids **1**–**4** with upconverting nanoparticles NaYF_4_: Yb^3+^: Er/NaYF_4_ in 0.9% sodium chloride solution was injected intravenously through the retroorbital sinus at a dose of 0.12 mg / mouse (5 mg/kg). Blood sampling was carried out from the tail vein after 1, 3, 5, 10, 30, 60, 120, 180 min and 1 day with tail trimming. Each blood sample was examined on an upconversion fluorescence microscope at four random locations. The corresponding photographs were taken and the number of nanoparticles was manually counted.

### 3.5. Particle Analysis

The size and the zeta-potential of the nanoparticles coated with lipid **4** were evaluated using a Zetasizer Nano ZS system (Malvern Panalytica, Malvern, UK) at 25 °C in deionized water (size) and in 10 mM NaCl solution (zeta-potential).

### 3.6. Cytotoxicity of Nanoparticles

Human Bj-5ta fibroblasts were seeded in a 96-well plate (5 × 10^3^ cells per well) followed by overnight incubation. Appropriate working suspensions of the nanoparticles coated with PEG-lipid **4** in full DMEM medium were prepared immediately prior to the experiments and added to the cells. Cell viability was determined by MTT assay after 24 h incubation. Cells were treated with MTT solution (0.5 mg/mL in DMEM) for 2 h, and the formed formazan crystals were dissolved in DMSO (100 µL). The optical density was measured at 570 nm usingVarioskan Flash reader (Thermo Scientific, Waltham, MA, USA). Cells without any treatment were used as controls (100%). 

## 4. Conclusions

In summary, we synthesized new PEG-glycerolipids **2**–**4** and demonstrated that the upconverting particles modified with lipid **4** formed stable dispersions in water and are nontoxic for fibroblasts. In vivo experiments showed that nanoparticles modified with lipid **4** were protected from exposure to blood serum proteins, maintaining the original nanoparticles’ fluorescence intensity and increasing circulation time in the bloodstream up to 3 h. 

## Data Availability

Not applicable.

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
