# Peer review of "New Cysteine-Containing PEG-Glycerolipid Increases the Bloodstream Circulation Time of Upconverting Nanoparticles"

_molecules, 2022, doi:10.3390/molecules27092763_

Round 1
Reviewer 1 Report
Nikolaeva et al. describe the synthesis of different PEGylated upconverting nanoparticles for enhanced in vivo circulation times. The presented manuscript is well-written and the results are mostly understandably presented. Nevertheless, there are several points that the authors should consider before I can suggest the work for publication:
- More details on the synthesis results are needed, such as NMR or FTIR spectra, especially in the main text. After all, this is the main point of the manuscript.
- More details on particle characterization: success of synthesis and PEGylation. This includes but is not limited to XRD, TEM, fluorescence spectrometry, or DLS.
- More data on in vivo studies, if available. Since the authors already have conducted the in vivo studies it is unfortunate that they did not provide more data from these. For example information of particle biodistribution or stability as a function of the polymer functionalization would have been interesting.
- Please provide the ethics approval for the conducted in vivo studies. Unless, I have missed it.
- Could you quantify the obtained fluorescence signal? How does it or the in vivo circulation time in general compare to literature?
- Page 1: line 30: re-phrase this too long and unclear sentence.
In summary, the presented manuscript lacks some crucial detailed data to fully support the made conclusions. This is especially valid for the particle/polymer synthesis and the in vivo studies.
Author Response
Dear Reviewer,
Thank you very much for the profound analysis of our manuscript. According to your comments, we carefully revised the article.
- More details on the synthesis such as NMR spectra.
We took into account your comment and included NMR spectroscopy data in the text of the article.
PEG-lipid 3 was synthesized from rac-1,2-di-O-octadecylglycerol 6, which was treated with imidazole complex with phosphorus (III) chloride (Scheme 2). The obtained product 7 was then activated with pivaloyl chloride and condensed with methoxy-PEG2000 following by oxidation with iodine. After isolation by means of column chromatography compound 3 was obtained with 38% yield over the two stages. The characteristic proton signals of glycerol CH2O-group (dd, 4.22-4.10 ppm) and PEG residue (br.s, 3.65 ppm) were found in the 1H NMR spectrum of lipid 3.
For synthesis of PEG-lipid 4 containing a cysteine residue, the initial rac-1,2-di-O-octadecylglycerol 6 was treated with N-bromosuccinimide and the resulting bromo derivative 8 [18] was condensed with Boc-L-cysteine methyl ester in the presence of a catalytic amount of KI (Scheme 2), giving compound 9 with 81% yield. Signals of L-cysteine protons (dt, 4.40 ppm, J 5.5, 7.9 Hz, CH-Cys and m, 2.85 – 2.81 ppm, SCH2-Cys) and Boc-group (s, 1.41 ppm С(Me)3) were found the 1H NMR spectra. Treatment of compound 9 with HCl/MeOH gave product 10, which was condensed with succinic anhydride to give compound 11 with 88% yield. Activation of the carboxyl group of compound 11 with N‑hydroxysuccinimide and subsequent condensation with amino-methoxy-PEG2000 gave PEG-lipid 12 with 63% yield. In the 1H NMR spectra of compounds 12 the proton signals of PEG residue as well as L-cysteine and glycerol (d 3.90 – 3.41 ppm, CH2O, 2 OCH2CH2, OCH3-Cys, CH2OCH3, PEG, CH2S) confirmed the formation of the desired product. To obtain the final ionic form of lipid 4, compound 12 was treated successively with lithium hydroxide and sodium hydrogen carbonate, which produced lipid 4 with 61% yield over the two stages.
- More details on particle characterization.
The particle size and zeta-potential upconverting nanoparticles coated with PEG-lipid 4 were evaluated. In addition, the conversion coefficient for untreated nanoparticle was specified.
Upconverting nanoparticles can be produced by various methods [19, 20]. We have developed a method of synthesis of upconverting nanoparticles that makes it possible to obtain monodisperse particles of NaYF4:M (М=Yb3+:Er3+) with a high conversion yield [21]. Mixture of the oxides Y2O3, Yb2O3, and Er2O3 were boiled in TFA-H2O (3:1 vol.), then CF3COONa was added and heated in a mixture of the high boiling-point solvents 1-octadecene and oleic acid (Fig. 2A). A special feature of this synthesis is a rapid temperature increase at the beginning of the synthesis (50 °C per minute) and a sharp temperature decrease (100 °C per minute) at the end of the synthesis to obtain monodisperse nanoparticles with a particle size of 75 nm ± 5 nm (Fig. 2B). The synthesis permit to obtain particles with a conversion coefficient (ratio of radiated power to absorbed power) of 14% at an excitation radiation intensity of 10 W/cm2.
Nanoparticles are rapidly removed from the bloodstream due to the reticuloendothelial system, followed by their accumulation in the spleen, the kidneys and the liver [22]. Particles coated with different shells reveal reduced adsorption by blood proteins and phagocytosis by macrophages, and characterized by prolonged circulation time in the organism [23]. To modify the surface of the upconverting nanoparticles, a solution of PEG-glycerolipids1-4 in dry chloroform was added to the nanoparticles. Chloroform was evaporated and a thin film was dried under 0.1 torr for 3 h and then suspended in water (MilliQ) to give a dispersion which was passed through a 0.2 μm membrane filter (PTFE, Merck). The dispersion was centrifuged at 6000 rpm for 1 h and nanoparticles modified with PEG-lipids 1-4 were redispersed in sterile water. Slightly opalescent dispersions were formed in the case of PEG-lipids 1, 3 and 4, which have ionizable groups, whereas in the case of the uncharged lipid 2 a non-transparent rapidly settling aqueous dispersion was obtained. The particle size and zeta-potential were evaluated using a Zetasizer Nano ZS (Malvern, USA) at 25 °C in deionized water (size) or in 10 mM NaCl solution (zeta-potential). Upconverting nanoparticles coated byPEG-lipid 4 had a size 165 ± 62 nm and a zeta potential -23.1 mV (see suppl. information).
- More data on in vivo
MTT-test was carried out for nanoparticles coated with PEG-lipid 4, the results was inserted in the text.
The nanoparticles coated with PEG-lipid 4 were non-toxic for Bj‑5ta fibroblasts after 24 h incubation (Fig. 2F), and the cell death measured by MTT assay did not exceed 10%. Thus, these upconverting nanoparticles were well tolerated by the cells and could be considered as promising candidates for in vivo applications.
The viability of Bj‑5ta fibroblasts after 24h incubation with nanoparticles, MTT assay. Data are the mean ± SD, cell viability of control (intact) cells is taken as 100% (E).
- Ethics approval.
All animal experiments were performed in accordance with European and Russian national guidelines for animal experimentation and were approved by the animal and ethics review committee of the Blokhin National Medical Research Center of Oncology, reference number 2017-034.
- How does it or the in vivo circulation time in general compare to literature? Quantify the obtained fluorescence signal.
Nanoparticles are rapidly removed from the bloodstream due to the reticuloendothelial system, followed by their accumulation in the spleen, the kidneys and the liver [22]. Particles coated with different shells reveal reduced adsorption by blood proteins and phagocytosis by macrophages, and characterized by prolonged circulation time in the organism [23].
. The signal from the nanoparticles modified by ionizable lipid 3 was observed up to 1.5 h whereas nanoparticles modified with cysteine-containing PEG-lipid 4 were visible for more than 3 h, more than 15 fold longer than the result obtained for the commercially available PEG-lipid 1. In addition, the circulation time of nanoparticles coated with PEG-lipid 4 was more than 3 fold longer than the time of other upconverting nanoparticles with different shells described in the literature [24, 25]. Analysis of the microphotographs using ImageJ (NIH, USA) software shown that the nanoparticles number was reduced from 260 to 18 over 10 min (Fig. 2D), whereas only moderate decrease of fluorescence intensity was observed (Fig. 2E).
|
|
||||
- Page 1: line 30: re-phrase this too long and unclear sentence.
Abstract: Upconverting nanoparticles have unique spectral and photophysical properties that make them suitable for development of theranostics for imaging and treating large and deep-seated tumors. Nanoparticles based on NaYF4 crystals doped with the lanthanides Yb3+ and Er3+ were obtained by high-temperature decomposition of trifluoroacetates in oleic acid and 1-octadecene. Such particles have pronounced hydrophobic properties. Therefore, to obtain stable dispersions in aqueous media for the study of their properties in vivo and in vitro, polyethylene glycol (PEG)-glycerolipids of various structures were obtained. To increase the circulation time of PEG-lipid coated nanoparticles in the bloodstream long-chain substituents are needed to be attached to the glycerol backbone using ether bonds. To prevent nanoparticle aggregation an L-cysteine derived negatively charged carboxy group should be included in the lipid molecule.

Reviewer 2 Report
The authors synthesized new cysteine-containing PEG-glycerolipid that increases the blood-2 stream circulation time of upconverting nanoparticles up to 3 hours. What I would like to know is how biocompatible this new substance is and whether to perform a cytotoxicity assessment. The authors should provide the above information if possible.
Author Response
Dear Reviewer,
Thank you very much for the profound analysis of our manuscript. According to your comments we carefully revised the article.
The authors synthesized new cysteine-containing PEG-glycerolipid that increases the blood-2 stream circulation time of upconverting nanoparticles up to 3 hours. What I would like to know is how biocompatible this new substance is and whether to perform a cytotoxicity assessment. The authors should provide the above information if possible.
MTT-test was carried out for nanoparticles coated with PEG-lipid 4, the results was inserted in the text.
The nanoparticles coated with PEG-lipid 4 were non-toxic for Bj‑5ta fibroblasts after 24 h incubation (Fig. 2F), and the cell death measured by MTT assay did not exceed 10%. Thus, these upconverting nanoparticles were well tolerated by the cells and could be considered as promising candidates for in vivo applications.
The viability of Bj‑5ta fibroblasts after 24h incubation with nanoparticles, MTT assay. Data are the mean ± SD, cell viability of control (intact) cells is taken as 100% (E).
